# Effect of Antirotational Two-Piece Titanium Base on the Vertical Misfit, Fatigue Behavior, Stress Concentration, and Fracture Load of Implant-Supported Zirconia Crowns

**DOI:** 10.3390/ma16134848

**Published:** 2023-07-06

**Authors:** Dario Adolfi, Manassés Tercio Vieira Grangeiro, Pietro Ausiello, Marco Antonio Bottino, João Paulo Mendes Tribst

**Affiliations:** 1Department of Dental Materials and Prosthodontics, Institute of Science and Technology, São Paulo State University (UNESP), São José Dos Campos 12220-000, Brazil; dario.adolfi@spazioodontologico.com (D.A.); manasses.grangeiro@unesp.br (M.T.V.G.); marco.bottino@unesp.br (M.A.B.); 2School of Dentistry, University of Naples Federico II, Via S. Pansini 5, 80131 Naples, Italy; 3Department of Reconstructive Oral Care, Academic Centre for Dentistry Amsterdam (ACTA), Universiteit van Amsterdam and Vrije Universiteit, 1081 LA Amsterdam, The Netherlands; j.p.mendes.tribst@acta.nl

**Keywords:** dental implants, dental implant abutment design, fatigue, finite element analysis, zirconia

## Abstract

This study investigated the effects of antirotational titanium bases on the mechanical behavior of CAD/CAM titanium bases used for implant-supported prostheses. The aim was to assess the impact on the marginal fit, fatigue behavior, stress concentration, and fracture load of implant-supported CAD/CAM zirconia crowns. Forty titanium implants were divided into two groups: those with antirotational titanium bases (ARs) and those with rotational titanium bases (RTs). Torque loosening and vertical misfit were evaluated before and after cyclic fatigue testing (200 N, 2 Hz, 2 × 10^6^ cycles). Fracture resistance was assessed using a universal testing machine (1 mm/min, 1000 kgf), and failed specimens were examined with microscopy. Three-dimensional models were created, and FEA was used to calculate stress. Statistical analysis was performed on the in vitro test data using two-way analysis of variance and Tukey’s test (α = 0.5). Results show that the presence of an antirotational feature between the implant and titanium base reduced preload loss and stress concentration compared to rotational titanium bases. However, there were no differences in vertical misfit and resistance to compressive load.

## 1. Introduction

The replacement of missing teeth with dental implants is a well-documented and widely accepted treatment option known for its high success rate [1,2,3]. As patients’ aesthetic demands have increased, there has been a need for the development of new implant designs [4], advancements in surgical techniques [5], and improvements in prosthetic components [6,7,8,9,10]. Achieving precise adaptation between the abutment/implant and abutment/prosthesis is crucial for ensuring the long-term success of the dental treatment [9,10]. However, mechanical complications, such as screw loosening and fractures of screws, implants, or components, continue to be significant factors affecting implant therapy [1,2,3,4,5,6,7,8,9].

In oral rehabilitations involving implants, mechanical complications more commonly arise from issues related to prostheses and their components [10,11,12]. Among implant-supported prostheses, screw loosening is the most frequently encountered complication, particularly in single-unit implant-supported crowns [1,10,12]. Additionally, over the long term, thread deformations in screws can occur, leading to a decrease in the ability to securely join the components and sometimes requiring the replacement of such structures [13,14,15].

Mechanical failure of the implant-supported structure can lead to biological complications such as inflammation of the peri-implant tissue, the formation of fistulas, and bone loss [1,11]. Therefore, long-term stability between the implant and the prosthetic component relies on maintaining proper preload force [9,12].

With the introduction of CAD/CAM systems, it is now possible to produce prosthetic components in a customized manner to recreate an appropriate emergence profile, facilitating the formation of the anatomical mucosal topography and cervical contour of the restoration [8,14]. CAD software utilizes scanned data from the patient’s models, which are then sent to a milling machine to produce a prosthetic component from metal or ceramic, eliminating the inherent inaccuracies associated with the lost-wax method. Recent analyses of the mechanical performance of zirconia abutments have shown they have lower fracture resistance compared to titanium abutments [15,16,17]. Zirconia exhibits promising industrial potential due to its high hardness, thermochemical stability, and biocompatibility. It finds extensive use in oral repair, tissue engineering, electrolyte materials, and structural applications. However, zirconia face issues like blockage, detachment, and cracking in high-stress environments [17,18]. To overcome this limitation, a hybrid component has been developed, featuring a titanium base (connection link) bonded to a mesostructure [19,20]. This hybrid abutment has showed superior aesthetics and adequate mechanical response in immediate- and long-term studies [10,17].

Hybrid abutments are available for external hexagon, internal hexagon, Morse taper, and alternative prosthetic connections [18]. It can be used at different heights and angulations, indicated for anterior and posterior crowns [19,20,21,22,23]. However, how the geometrical morphology of the titanium base, including antirotational and conical designs, affects the stability of the implant–abutment connection has not been investigated yet [22,24]. Therefore, the present study aimed to assess the effect of an antirotational titanium base on the marginal fit, fatigue behavior, stress concentration, and fracture load of implant-supported CAD/CAM zirconia crowns.

## 2. Materials and Methods

The materials that were used in the present study are described in Table 1.

### 2.1. Specimen Preparation

This study used forty (40) implants (IPX 4012, post-extraction internal hexagonal connection, 4 × 12 mm, 4 mm platform, Nueva Galimplant, Sarria, Lugo, Spain), with a Morse taper connection [25]. These implants were installed in resin cylinders obtained through milling, with the following dimensions: 22 mm in height and 25 mm in diameter. For implant insertion, centered perforations were made in the resin cylinders using a pilot drill, followed by a drill size 2. The implants were placed inside each resin cylinder using an implant insertion wrench, ensuring that the maximum torque did not exceed 60 N, and the platform was positioned 3 mm above the resin, following the ISO 14801 second edition of 2007 [21,22,23,24,25,26,27,28].

Half of the implants received the antirotational two-piece titanium bases (ARs), and the other half received the rotational two-piece titanium bases (RTs). The fixation screws were inserted into their central perforations and tightened with a manual torque wrench until properly seated. The maximum defined preload was set at 20 N·cm using a digital torque wrench.

An acrylic dental mannequin was used to create the provisional restoration, which served as a base for scanning and fabricating all the milled zirconia restorations used in this study. The region of tooth 21 was used to position the implant and simulate the ideal three-dimensional position on the working model. The acrylic tooth 21 was separated 2 mm below the cervical contour line and used to produce the provisional restoration over the implant.

The provisional restoration was scanned using a CEREC scanner (Dentsply Sirona Dental Systems, Bensheim, Germany). The captured images were exported in STL format and transferred to the Ceramill Mind software, ver 4.0 (AmannGirrbach, Koblach, Austria) to replicate the exact shape of the restoration. With the project fully defined, the machining stage started. The project data were sent to the milling center (Ceramill Motion 2, AmannGirrbach, Koblach, Austria), which milled 40 restorations from a zirconia disc with identical anatomical interfaces for the multi-position CEREC aesthetic system containing a screw-access channel.

All restorations were milled with 20% larger dimensions to compensate for the volumetric shrinkage that occurs during the zirconia sintering process. After milling, all restorations had the same morphology and size, and were separated from the block using a low-speed carbide bur. Excess was removed with a diamond bur. The restorations were then polished with low-speed abrasive rubber before the sintering process to ensure a smooth surface. After polishing, all specimens were cleaned with isopropyl alcohol in an ultrasonic bath for 5 min.

The sintering of the restorations was carried out using a high-temperature sintering oven (InFire HTC Speed, Dentsply Sirona Dental Systems, Bensheim, Germany). The restorations were placed inside the firing tray and completely dried to enable the use of the accelerated firing cycle. The multi-position interfaces for the CEREC system were tightened onto the implant with a positioned base, which served as a two-piece titanium base. The screw heads were then protected with Teflon tape. The external surfaces of the interfaces were sandblasted with 50 μm aluminum oxide (Al_2_O_3_) at a maximum pressure of 2.5 bar for 10 s at a distance of 10 mm and cleaned with isopropyl alcohol in an ultrasonic bath for 5 min. The silane agent (Clearfill Ceramic Primer, Kuraray Medical Inc., Okayama, Japan) was applied inside the zirconia restorations for 60 s, followed by an air jet to remove the excess. Subsequently, the metal primer (Alloy Primer, Kuraray Medical Inc., Okayama, Japan) was applied to the abutment surface for approximately 5 s (Figure 1).

The resin cement (Panavia F2.0 Kuraray Medical Inc., Okayama, Japan) was manipulated by mixing equal amounts of pastes A and B for 20 s. Subsequently, the cement was applied to the outer surrounding walls of the interfaces and the inner walls of the restorations. The restorations were held under digital pressure, and excess cement was removed with a microbrush. Oxygen-blocking gel (Oxyguard Kuraray Medical Inc., Okayama, Japan) was applied to the bonding area between the interfaces and the zirconia restorations. Light curing was performed using an LED unit for 20 s on each surface of the restorations (high intensity of 1000 mW/cm^2^, with wavelengths ranging from 395 to 480 nm—Valo, Ultradent Products, South Jordan, UT, USA). Cement residues were removed with polishing cups, and the specimens were stored in distilled water for 48 h for complete resin curing.

### 2.2. Vertical Misfit

As a baseline, immediately after the storage period, the interfaces between the titanium bases and the restorations of each specimen were analyzed using an optical microscope (Discovery V20, Zeiss, Jena, Germany) at a magnification of 40×, and the vertical misfit was measured in micrometers. Ten measurements of the interfaces were taken on each side of the restorations (4 regions (buccal, mesial, distal, and palatal)) by a single trained examiner [29]. After fatigue cycling, the same protocol was applied to measure the vertical misfit in the long-term simulation.

### 2.3. Capacity to Maintain Placement Torque

The prosthetic screws were tightened to 20 N·cm using a digital torque wrench (TQ 680; Instrutherm Measurement Instruments, São Paulo, Spain). The removal torques of the screws were measured after 5 min (initial removal torque of preload) using the same digital torque wrench. The data were collected, and the torque was reapplied to the screws.

The preload efficiency for each abutment was calculated based on the following formula: preload efficiency (%) = removal torque/tightening torque × 100 [10].

### 2.4. Mechanical Fatigue

The mechanical fatigue test was conducted using a thermo-mechanical cyclic loading device. For this purpose, forty specimens (n = 20 per group) were placed in a stainless-steel base with a 30-degree angulation relative to the ground, following ISO 14801, to assess the aging effect. Subsequently, the specimens were subjected to a load of 200 N at a frequency of 2 Hz for 2 × 10^6^ cycles (Figure 2) [10,29].

All aged specimens were re-evaluated for vertical misfit and torque maintenance capacity to assess the effect of cyclic fatigue on these parameters [10].

### 2.5. Post-Fatigue Fracture Load

Subsequently, the specimens were repositioned to their initial condition and subjected to the compressive load-to-failure test using a universal testing machine (EMIC DL 1000, São José dos Pinhais, Brazil). Compressive load will be applied to each specimen through a unidirectional vertical platform at a rate of 0.5 mm/min until failure, defined as either screw or implant–abutment interface fracture. The maximum load at failure will be recorded in Newtons [10]. Each fractured restoration was visually inspected at 25× magnification under an optical microscope (Zeiss Discovery V20; LLC, Oberkochen, Baden-Württemberg, Germany).

### 2.6. Finite Element Analysis

Using computer-aided design software (Rhinoceros version 5.0 SR8, McNeel North America, Seattle, WA, USA), a fixation cylinder model was recreated for an in vitro study, maintaining dimensions of 25 × 20 mm. Implant designs (4.3 × 11.5 mm) were then drawn based on 4.3 mm diameter circles to determine the implant’s three-dimensional structure [18]. The models were created following the BioCAD protocol using STL files provided by Nueva Galimplant (Sarria, Lugo, Spain). After converting the implant’s geometry into a solid, including surface and polysurface union, the models consisted of a fixation cylinder, prosthetic screw, titanium base, and prosthetic restoration [10]. The final models were exported as STEP files after thorough verification (Figure 3).

Each previously described geometry was exported to the computer-aided engineering software (ANSYS 19.2, ANSYS Inc., Houston, TX, USA). A static structural analysis was conducted to calculate the results. In the mechanical module (Table 2), material information was assigned to each solid component considering them isotropic and homogeneous, utilizing the elasticity modulus and Poisson ratio from previous studies [10,30,31,32,33]. Two different abutment designs were considered, similar to the in vitro test (Figure 4 and Figure 5).

The contacts were considered bonded, except between the metallic structures, where a friction coefficient of 0.3 μ was assigned between the bodies. The number of tangent faces between the solids was equalized. An initial division with tetrahedral elements was automatically generated.

Subsequently, a 10% convergence test was employed to assist in mesh refinement and control, ensuring minimal influence on the results of the mathematical calculations [10]. The mesh convergence was applied based on von Mises stress peaks, generated at the abutment interface, and collected by the Max probe from the mechanical module. The mesh density was adjusted by refining the existing mesh, locally in critical regions, using h-refinement (decreasing element size uniformly). TET10 elements were used in both models, with an element size of 0.4 mm, totaling 158,764 elements with 276,653 nodes for the ARs model and 158,310 elements with 274,785 nodes for the RTs model. The average aspect ratio was 1.72 with a maximum element skewness of 0.45 in the entire volume of the element model.

The loading was performed in the incisal region of the restoration as per ISO 14801. The fixation location was defined beneath the resin cylinder surface, simulating specimen support on a plane. An oblique load (Figure 6) was applied to the palatal surface (45°, 150 N) [27]. The prosthetic screw was preloaded to simulate the tightening corresponding to the torque applied in the laboratory model (20 N·cm).

The requested solutions were sought in terms of Von Mises stress for the implant and abutment in each group. The results are presented in stress maps with identical scales for visual comparison, and absolute values were used for quantitative analysis of stress peaks.

### 2.7. Statistical Analysis

For both the ARs and RTs groups, the vertical misfit results were submitted to a general linear model for two-way analysis of variance (ANOVA) considering the following factors: “Region” (buccal, mesial, distal, and palatal) and “aging” (Yes or No). In the sequence, vertical misfit and torque maintenance capacity results were submitted to a general linear model for two-way analysis of variance (ANOVA) considering the following factors: “abutment type” (ARs or RTs) and “aging” (Yes or No). The load to failure was submitted to one-way ANOVA considering the “abutment type” factor. The Tukey test was used to evaluate the comparisons between groups. All tests presented α value of 0.5. The stress maps were analyzed qualitatively and quantitatively with the stress peaks in the restoration, titanium base, and prosthetic screw.

## 3. Results

### 3.1. Measurement of Vertical Misfit

For the vertical misfit evaluation in the ARs, the average values ranged from 95.61 to 99.91 μm. According to the statistical test, the crown region was not significant (F-value = 0.13; *p*-value = 0.94); nor were the aging (F-value = 0.01; *p*-value = 0.96) (Table 3) or the interaction of both factors (F-value = 0.08; *p*-value = 0.97).

The vertical misfit evaluation in the RTs group showed average values ranging from 96.12 to 104.30 μm. According to the statistical test, the crown region was not significant (F-value = 0.01; *p*-value = 0.99); nor were the aging (F-value = 0.15; *p*-value = 0.69) (Table 4) or the interaction of both factors (F-value = 0.00; *p*-value = 0.99).

A second two-way ANOVA was performed to compare the vertical misfit between the RTs and ARs regardless of the evaluated crown’s region. Similarly to the previous data, according to the statistical test, the misfit between both connection types was similar (F-value = 0.20; *p*-value = 0.65), without an effect due to the simulated aging (F-value = 0.08; *p*-value = 0.71) or the interaction of both factors (F-value = 0.12; *p*-value = 0.73). The average vertical misfit for the ARs was 95.25 ± 44.96 before and 98.23 ± 48.94 after aging, while the RTs showed 98.98 ± 31.88 before and 98.75 ± 41.58 after aging.

### 3.2. Torque Maintenance Capacity

Regarding torque maintenance capacity (Table 5), the interaction of factors presented a significant effect (F-value = 4.28; *p*-value = 0.04) on the capacity of the abutment to maintain the torque. The fatigue cycling significantly decreased the preload efficiency (in 32.75%) for the RTs group, while the ARs showed a torque decrease of 39.75%. The difference between both evaluated connections after fatigue cycling was 7% (1.4 N·cm), being detected as significant.

### 3.3. Compression Test

One-way ANOVA revealed similar (F-value = 0.00; *p*-value = 0.94) mean values of fracture resistance for the ARs (554.25 ± 68.8 N) and RTs (552.89 ± 58.1 N). Altogether, 80% of the restorations using the ARs failed in the screw and 20% failed at the cervical level of the crown, while 70% of the restorations using the RTs failed in the screw and 30% failed in the crown (Figure 7).

### 3.4. FEA

Finite element simulation showed that both modalities presented similar stress distribution at the external surface of the crown and implant threads (Figure 8). In analyzing the titanium base surface, was noted that the ARs presented fewer red colors in the colorimetric map of stress, which means it experienced lower stress concentration than the RTs group (Figure 8).

On the other hand, for the implant, the ARs showed a similar stress pattern in the threads than the RT, with lower stress than the abutments of both models, indicating a lower probability of failure in this area. The stress peak results are summarized in Table 6.

## 4. Discussion

The issue of vertical misfit has long been recognized as a significant concern in implant dentistry. Vertical misfit refers to the discrepancy between the implant fixture and the abutment, which can lead to complications such as peri-implant bone loss, implant failure, and compromised esthetics [33,34,35]. The prosthetic–abutment vertical misfit in clinical settings usually ranges from 50 to 160 μm in implant-supported prostheses manufactured using the casting technique [36]. The present results show that both evaluated abutment designs showed acceptable values of misfit in the range of 100 μm, before and after fatigue cycling. Therefore, a rotational or antirotational titanium base can be indicated to minimize vertical misfit during implant restoration procedures through meticulous fabrication techniques and precise fitting of the abutments.

The choice of abutment material and design plays a vital role in maintaining the placement torque of dental implants. In the present investigation, both abutments were assessed according to their capacity to sustain placement torque over time [36,37]. It was observed that the antirotational titanium base demonstrated superior torque maintenance compared to those with a rotational feature. This is likely because the hexagonal design of the abutment acts as rotational resistance against moments [38]. It is important to notice that both abutments showed a significant preload reduction after fatigue. However, the difference of 1.4 N·cm between them is far from being clinically relevant in comparison with the torque loss caused by the screw elastic recovery, which can reach 50.71% depending on the abutment design [39]. Therefore, the RTs group showed an acceptable behavior that can be associated with fully tapered internal connections, since the screw tightening is not fully transferred to preload because of its unique design. In summary, it will cause a wedging effect through the settlement of the abutment, increasing the frictional force between the implant–abutment joint [38]. These findings emphasize the importance of selecting appropriate abutments to ensure the long-term stability and functional success of dental implants [40]. Additionally, in this study, both the ARs and RTs groups were original components from the same manufacturer of the implant fixture. According to the literature, enhanced fit is expected when original components are used, and the original abutments exhibited lower percentages of torque reduction after cyclic loading than non-originals [40].

Implant mechanical fatigue and fracture remain significant concerns in implant dentistry [41,42]. Furthermore, implants with higher fracture loads are associated with enhanced durability and reduced risk of failure [43]. These findings highlight the importance of selecting implant systems with superior mechanical properties to optimize long-term clinical outcomes and minimize the occurrence of implant fractures. In the present study, both groups showed similar post-fatigue fracture load, with values higher than 500 N. It has been reported that the maximum bite force in the anterior region can range from 100 to 300 Newtons (N), depending on various factors, such as gender, age, occlusal condition, and method of measurement [44,45,46]. Therefore, both designs are suitable for use in the simulated clinical condition.

Understanding the stress distribution within dental implant restorations is critical for evaluating their biomechanical behavior and predicting potential complications. The complementary FEA was used to evaluate whether either abutment design exhibited any substantial differences in stress distribution that could potentially affect the long-term stability of implant-supported restorations. The stress maps revealed that both abutment designs demonstrated similar stress patterns, with only minor differences observed at the connection level. The stress region is correspondent to previous studies that evaluated different implants designs and brands, but with similar Morse taper connection [47,48,49].

This suggests that the overall stress distribution within the restorations was comparable between the two designs. These small differences at the connection level may be attributed to variations in the geometric features and differences in the surface contact of the ARs and RTs with the implant. Nonetheless, these differences were not significant enough to influence the biomechanical behavior of the restorations or raise concerns regarding their structural integrity according to the other in vitro tests. Therefore, both abutment designs can be considered as viable options for implant-supported restorations in the anterior region, as they exhibited comparable stress patterns and are likely to withstand the maximum bite forces encountered in this area.

Finite element analysis (FEA) is a powerful computational tool used to analyze complex structural behavior [50,51]. However, it has certain limitations that must be considered when extrapolating its results to a clinical scenario. Firstly, FEA heavily relies on accurate material properties and boundary conditions, which can be challenging to determine for biological tissues [52,53,54,55]. Additionally, FEA models typically consider static conditions and neglect dynamic factors such as tissue deformation and physiological loads. Correlating in vitro experiments with FEA can enhance the reliability of the present results. However, it is important to note that even with this correlation, there are still limitations to consider, such as the complexity of oral tissues and the dynamic nature of fatigue processes. Therefore, it is still crucial to exercise caution and consider additional factors when extrapolating results to clinical scenarios [56,57,58,59,60].

## 5. Conclusions

The clinical significance of incorporating an antirotational feature between the implant and titanium base is two-fold: firstly, it leads to a reduction in preload loss and stress concentration, promoting greater stability and longevity of the implant-supported prosthesis. This minimizes the risk of complications associated with implant loosening or failure. Secondly, while no significant differences were found in vertical misfit and resistance to compressive load, these factors remain crucial considerations for successful restorations. It can be suggested that both antirotational and rotational titanium bases may be considered for use in implant-supported prostheses. Clinicians can choose between these options based on individual patient needs, implant stability, and other clinical factors.

## Figures and Tables

**Figure 1 materials-16-04848-f001:**
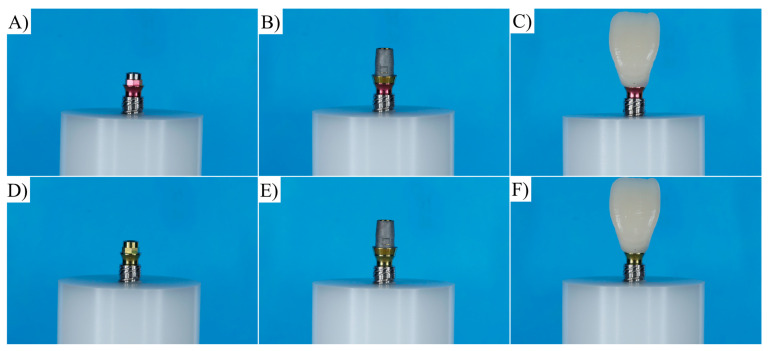
Specimen preparation for ARs group (**A**–**C**) and RTs group (**D**–**F**).

**Figure 2 materials-16-04848-f002:**
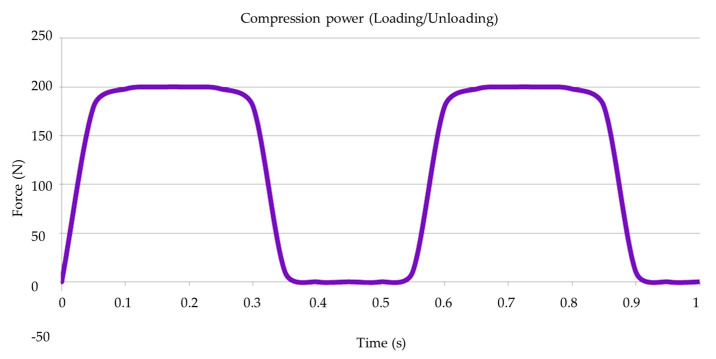
Graph of loading/unloading during fatigue cycling.

**Figure 3 materials-16-04848-f003:**
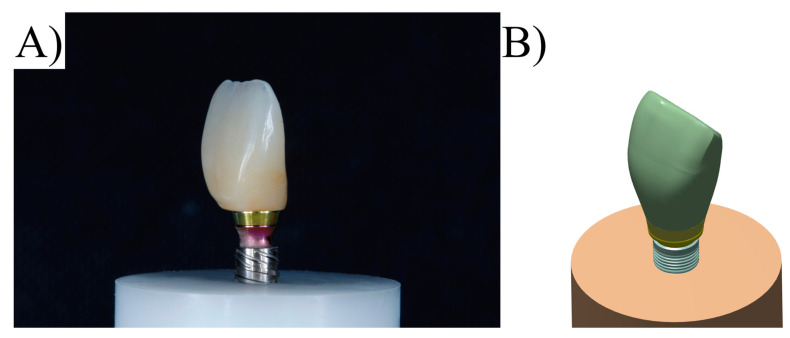
(**A**) Final setup of experimental specimen and (**B**) CAD design with similar geometry for numerical analysis.

**Figure 4 materials-16-04848-f004:**
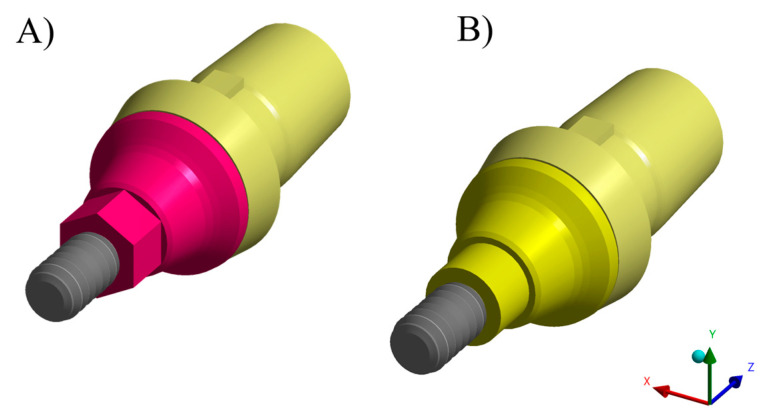
Different 3D designs of titanium bases evaluated. (**A**) Antirotational base and (**B**) rotational base.

**Figure 5 materials-16-04848-f005:**
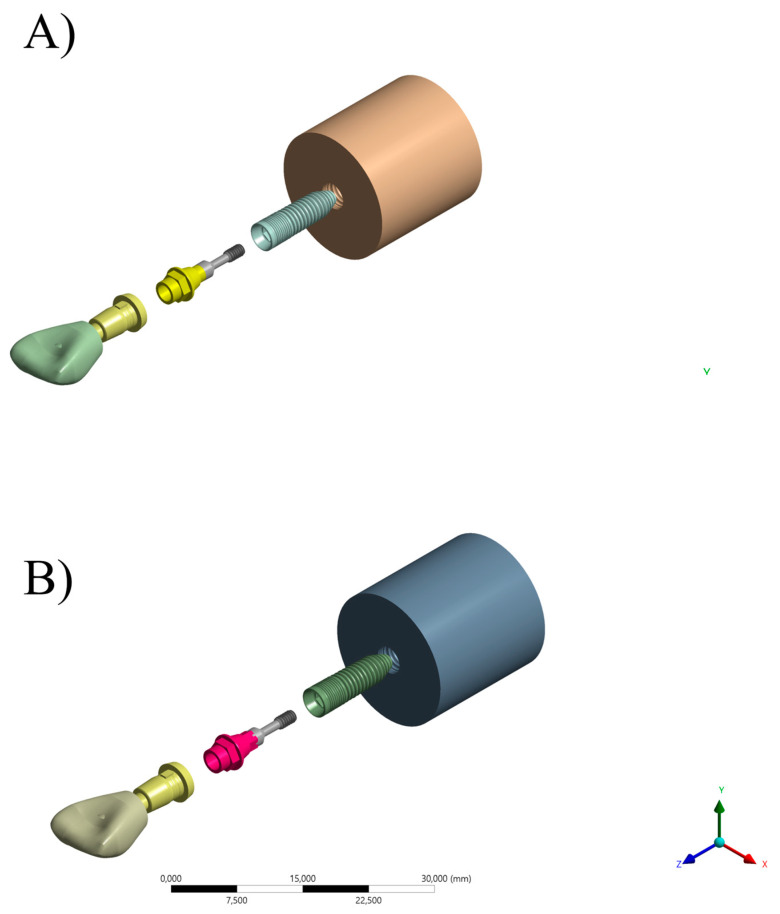
Isometric with disassembled view geometries for both 3D designs of titanium bases evaluated. (**A**) Rotational base and (**B**) antirotational base.

**Figure 6 materials-16-04848-f006:**
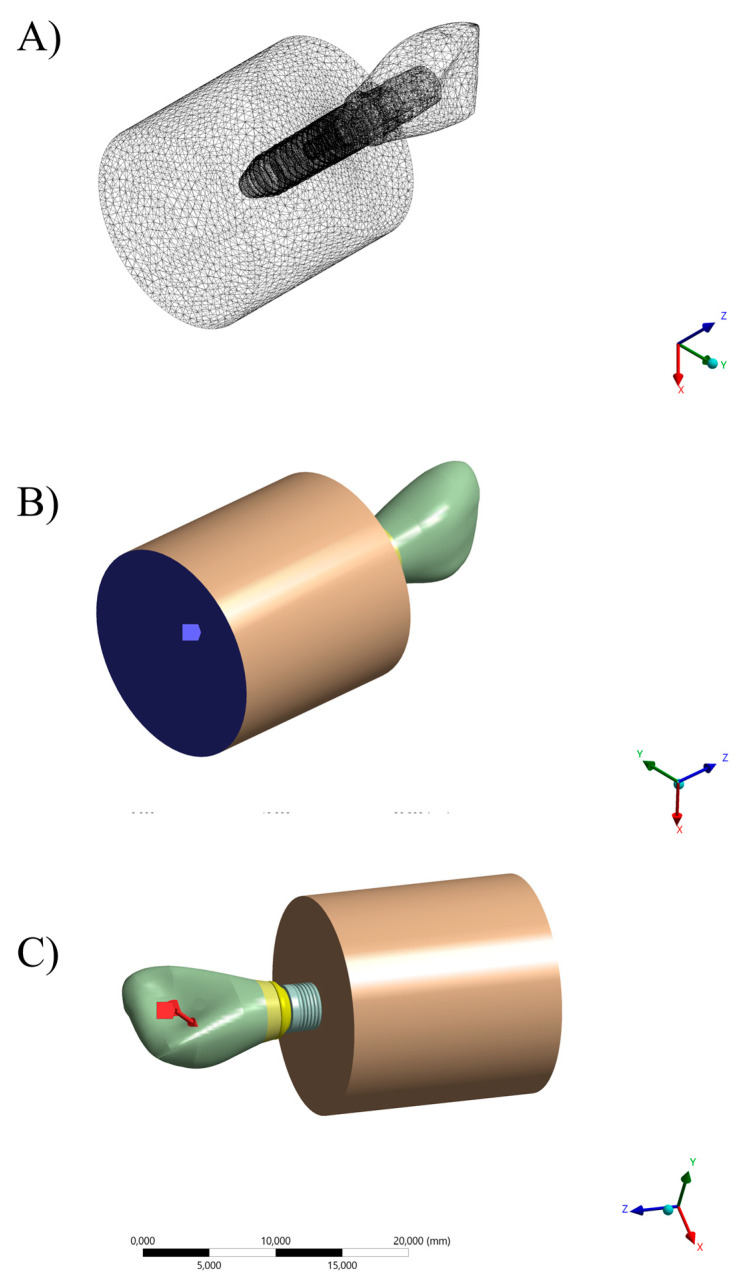
(**A**) Finite element model after meshing refinement and boundary conditions applied in the simulation, (**B**) fixed support, and (**C**) loading region at palatal surface.

**Figure 7 materials-16-04848-f007:**
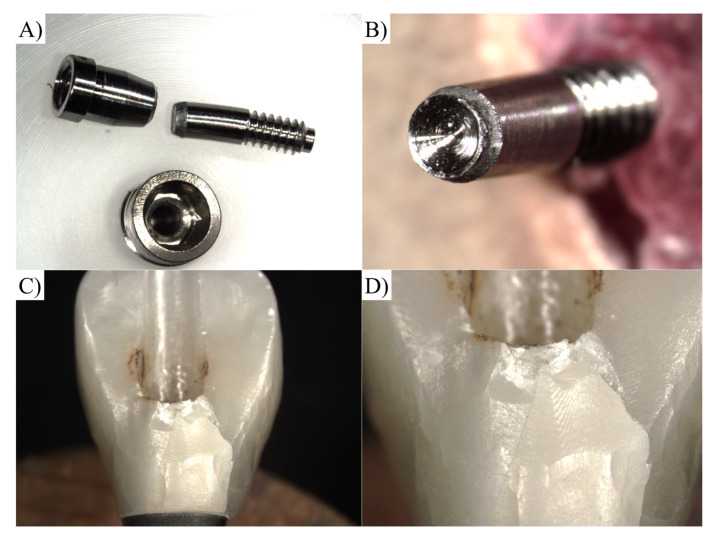
Representative failure mode during compression test. (**A**,**B**) Screw fracture at the screw head; (**C**,**D**) fracture of the zirconia crown.

**Figure 8 materials-16-04848-f008:**
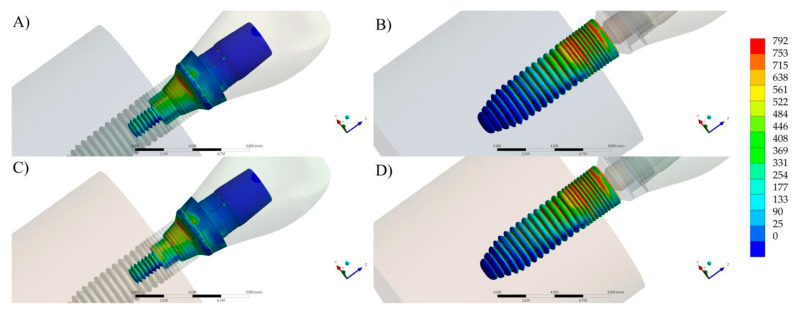
The von Mises stress distribution according to each evaluated abutment design. (**A**,**B**) Stress maps for abutment and implant when using the ARs abutment. (**C**,**D**) Stress maps for abutment and implant when using RTs abutment.

**Table 1 materials-16-04848-t001:** Trademarks and manufacturers of the materials that were used in the present study.

Material	Commercial Name	Manufacturer
Titanium	Implant postextracción conexión interna hexagonal 4 × 12 mm platform 4 mm (batch 202009029) Cod IPX 4012	New Galimplant, Sarria, Lugo, Spain
Titanium	Abutment multi-posicíon recto anti-rotacional solidario altura 2 mm CI hexagonal (batch 202010078) Cod KMUSA S04020	
Titanium	Abutment multi-posicíon recto anti-rotacional altura 2 mm CI hexagonal (batch 202005016) Cod KMUSA 04020	
Titanium	Interfase compatible con Sistema Cerec para multi-posicíon estético anti-rotacional (batch 202005001) Cod EPCERCMUA 40	
Self-curing acrylic resin	Unifast	GC America; Alsip, IL, USA
Inclusion resin	POM (Delrin) (batch 166976)	Dupont; Wilmington, DE, USA
Zirconia disk for mesostructure	3D pro ML (batch W200604ATB2M-01-P)	Aidite Technology Co., Ltd., Shenzhen, China
Cementing agent	Panavia F 2.0 Light (batch 000133)	Kuraray Noritake Dental Inc., Okayama, Japan
Primer for metal	Alloy Primer (batch AA0096)	Kuraray Noritake Dental Inc., Okayama, Japan
Primer for zirconia	Ceramic Primer Plus (batch 3P0053)	Kuraray Noritake Dental Inc., Okayama, Japan

**Table 2 materials-16-04848-t002:** Mechanical properties of the materials used in this study.

Material	Elastic Modulus (GPa)	Poisson Coefficient	Reference
Titanium	105	0.33	[32]
Zirconia	205	0.3	[30,31]
Fixation resin	2.8	0.3	[10]
Resin cement	11.8	0.3	[33]

**Table 3 materials-16-04848-t003:** Average vertical misfit (μm) according to each region of the crown, before and after aging for the ARs group.

Region	Aging	Marginal Misfit
Buccal	No	95.61 ± 45.21
	Yes	99.91 ± 50.38
Distal	No	94.84 ± 45.56
	Yes	97.24 ± 49.02
Palatal	No	96.07 ± 47.29
	Yes	98.24 ± 50.43
Mesial	No	94.47 ± 45.24
	Yes	97.52 ± 49.69

**Table 4 materials-16-04848-t004:** Average vertical misfit (μm) according to each region of the crown, before and after aging for RTs group.

Region	Aging	Marginal Misfit
Buccal	No	99.72 ± 32.3
	Yes	104.30 ± 55.06
Distal	No	97.95 ± 30.02
	Yes	96.12 ± 37.48
Palatal	No	99.33 ± 34.08
	Yes	96.57 ± 35.68
Mesial	No	98.92 ± 33.52
	Yes	98.02 ± 37.85

**Table 5 materials-16-04848-t005:** Removal torque (N·cm) according to group before and after aging. Grouping distribution according to Tukey test (95%).

Group*Aging	Torque (N·cm)	Grouping		
ARs*No	20.00	A		
RTs*No	20.00	A		
RTs*Yes	13.45 ± 2.18		B	
ARs*Yes	12.05 ± 2.08			C

**Table 6 materials-16-04848-t006:** The von Mises stress peak per model (MPa) in the abutment and implant structures.

Model	Structure	Stress Peak (MPa)
AR	Abutment	818.56
	Implant	754.35
RT	Abutment	994.36
	Implant	755.87

## Data Availability

Data available on request of the first author.

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
