# Peer review of "Effect of Antirotational Two-Piece Titanium Base on the Vertical Misfit, Fatigue Behavior, Stress Concentration, and Fracture Load of Implant-Supported Zirconia Crowns"

_materials, 2023, doi:10.3390/ma16134848_

Round 1

Reviewer 1 Report

The research topic focuses on the effect of pillar selection based on its geometry, anti-rotational two-piece, on the mechanical behavior that contributes to the long-term success of zirconia implant-supported prostheses. It is a relevant topic presented with a concise and detailed methodology following standardized protocols. The results and discussion are presented systematically. However, I suggest considering the following points:

(1) In line 15-16, the sentence should be completed. Can it vary considerably depending on...?

(2) In the results, specifically in the Compression Test (line 253-256), I assume there is an error in the designation, either for the RT or AR assembly.

(3) In line 282-284, this sentence should be exchanged with the position of lines 279-282 or removed, as the objective of the work at that point in the manuscript has been mentioned multiple times and is therefore known.

(4) In line 287-289, what other type of pillar structure or method is compared to conclude that the rotational or anti-rotational titanium base may be indicated to minimize vertical misfit? While they present a value within the reported range, there may be pillars with lower vertical misfit (50µm). This sentence needs to be rewritten to avoid misleading conclusions.

The references are appropriate and refer to highly relevant bibliographic sources. The tables and figures are clear and represent the data well.

The article is innovative, presents a good discussion of the results, and the introduction along with the methodology is appropriate. It is a good scientific work.

The article is innovative, presents a good discussion of the results, and the introduction along with the methodology is appropriate. It is a good scientific work.

Author Response

The research topic focuses on the effect of pillar selection based on its geometry, anti-rotational two-piece, on the mechanical behavior that contributes to the long-term success of zirconia implant-supported prostheses. It is a relevant topic presented with a concise and detailed methodology following standardized protocols. The results and discussion are presented systematically. However, I suggest considering the following points:

Dear reviewer, thank you for providing me with your valuable review suggestions. I appreciate your input and will take them into consideration to improve the present study.

 (1) In line 15-16, the sentence should be completed. Can it vary considerably depending on...?

The sentence has been removed from abstract section according to other reviewer request.

 (2) In the results, specifically in the Compression Test (line 253-256), I assume there is an error in the designation, either for the RT or AR assembly.

Thank you very much for noticing that. The sentence has been corrected.

 (3) In line 282-284, this sentence should be exchanged with the position of lines 279-282 or removed, as the objective of the work at that point in the manuscript has been mentioned multiple times and is therefore known.

The sentence has been removed as suggested.

 (4) In line 287-289, what other type of pillar structure or method is compared to conclude that the rotational or anti-rotational titanium base may be indicated to minimize vertical misfit? While they present a value within the reported range, there may be pillars with lower vertical misfit (50µm). This sentence needs to be rewritten to avoid misleading conclusions.

The text has been corrected as requested. The cited reference considered implant-supported prostheses manufactured by casting technique.

 The references are appropriate and refer to highly relevant bibliographic sources. The tables and figures are clear and represent the data well.

 The article is innovative, presents a good discussion of the results, and the introduction along with the methodology is appropriate. It is a good scientific work.

We appreciate.

Reviewer 2 Report

Please condense the abstract - in my opinion, it should be concise and indicate what will be in the manuscript, without unnecessary details - this is what the paper is supposed to be about.

The research material used in the laboratory should be called "specimen" in the paper - not "sample" - please correct the manuscript in this respect.

Please add nomenclature to the manuscript - a full list of symbols, abbreviations and markings. It may be at the end of the paper.

Short but to the point introduction.

Table 1 should be improved - more condensed and better formatted.

The paper clearly lacks technical drawings of the specimens used. Please show full isometric views, technical drawings with dimensions, information about the roughness of the speciemns - as we know it affects the results of fatigue tests - roughness affects the fatigue strength. The paper must necessarily include information about the specimens, technical drawings, dimensions of the specimens, illustrations with isometric views of the CAD models of the specimens used.

The manuscript should specify exactly how many specimens were subjected to fatigue tests, what signals were recorded, what these signals were used for. It is worth illustrating these studies with sample charts. In these tests, at least 5 specimens should be used - as it is known, fatigue tests are subject to a significant dispersion - scater of the results. It is worth showing identical graphs from fatigue tests - forces as a function of displacement or time for several specimens, in a specific period of time. In this regard, you should add some results, show their dispersion, perform an analysis.

Tables in the manuscript should be better presented, formatted and discussed.

In addition to photos of broken bolts, it is worth showing and measuring fractures using a laboratory microscope. Please complete your manuscript in this regard.

There is absolutely no information about FEM modeling in the manuscript. Here you should specify the solver, the finite elements used, the number of nodes and integration points in the finite element, the type of interpolation in the finite element, show the FEM model with assumed boundary conditions, external load, etc. Please indicate the material model used in the calculations, show its stress - strain curve give its mechanical parameters. It is necessary to show what type of finite elements were, what was their size - also relate it to the geometric parameter of the specimens, etc. In this respect, the manuscript does not meet any requirements - it disqualifies it from publication.

Please also tell me how the convergence of the numerical model was determined - as it was tested - this is also not in the paper.

The great deficiencies in the paper suggest that it should be rejected, but I believe that the authors should be given a chance to revise it and resubmit it for review. I suggest a major revision.

Minor editing of English language required.

Author Response

Please condense the abstract - in my opinion, it should be concise and indicate what will be in the manuscript, without unnecessary details - this is what the paper is supposed to be about.

The abstract has been shortened as requested. Maximum 200 words have been kept as informed by the guidelines.

The research material used in the laboratory should be called "specimen" in the paper - not "sample" - please correct the manuscript in this respect.

Corrected.

Please add nomenclature to the manuscript - a full list of symbols, abbreviations and markings. It may be at the end of the paper.

Thank you for this suggestion. As an optional feature, we decided not to include that. However all acronyms are identified in the main document.

Short but to the point introduction.

We appreciate.

Table 1 should be improved - more condensed and better formatted.

Table 1 has been improved as requested. The forma follows the journal’s guidelines.

The paper clearly lacks technical drawings of the specimens used. Please show full isometric views, technical drawings with dimensions, information about the roughness of the speciemns - as we know it affects the results of fatigue tests - roughness affects the fatigue strength. The paper must necessarily include information about the specimens, technical drawings, dimensions of the specimens, illustrations with isometric views of the CAD models of the specimens used.

We agree that  roughness affects the fatigue strength. Therefore the finishing protocol for each sample was glazing. The glaze application is the clinical protocol used for finishing dental ceramic restorations, reducing the roughness and improving the brightness. Unfortunately, in this study, surface roughness was not measured; however, the laboratory protocols were the same as those performed in the dental prosthesis laboratory. In this way, the final surface topography is similar to that used by the dentist when making this therapeutic modality. More information about the specimens has been added as well as the isometric view and dimensions.

The manuscript should specify exactly how many specimens were subjected to fatigue tests, what signals were recorded, what these signals were used for. It is worth illustrating these studies with sample charts. In these tests, at least 5 specimens should be used - as it is known, fatigue tests are subject to a significant dispersion - scater of the results. It is worth showing identical graphs from fatigue tests - forces as a function of displacement or time for several specimens, in a specific period of time. In this regard, you should add some results, show their dispersion, perform an analysis.

The topic 2.4. Mechanical fatigue has been corrected with the following sentence “For this purpose, forty specimens (n=20 per group) were placed in a stainless steel base” instead of “For this purpose, all specimens were placed in a stainless steel base”. About the dispersion from fatigue test, all specimens survived the fatigue cycling and were later submitted to the compressive load in a universal testing machine. The fracture origin can not be identified when applying such method. A graph showing the function of displacement on time was added according to the fatigue parameter simulated in this study. This protocol is similar to a previous investigation about titanium base design https://doi.org/10.11607/jomi.7731.

Tables in the manuscript should be better presented, formatted and discussed.

The presentation and formatting followed the journal’s guidelines. Actually the whole manuscript has been formatted using the provided template. If there are any other points that should be better discussed we would be glad to receive an more specific feedback.

In addition to photos of broken bolts, it is worth showing and measuring fractures using a laboratory microscope. Please complete your manuscript in this regard.

There is absolutely no information about FEM modeling in the manuscript. Here you should specify the solver, the finite elements used, the number of nodes and integration points in the finite element, the type of interpolation in the finite element, show the FEM model with assumed boundary conditions, external load, etc. Please indicate the material model used in the calculations, show its stress - strain curve give its mechanical parameters. It is necessary to show what type of finite elements were, what was their size - also relate it to the geometric parameter of the specimens, etc. In this respect, the manuscript does not meet any requirements - it disqualifies it from publication.

Topic 2.6 presents the modeling step in CAD “Using computer-aided design software (Rhinoceros version 5.0 SR8, McNeel North America, Seattle, WA, USA), a fixation cylinder model was recreated for an in vitro study, maintaining dimensions of 25 x 20 mm. Implant designs (4.3 x 11.5 mm) were then drawn based on 4.3 mm diameter circles to determine the implant's three-dimensional structure [14]. The models were created following the BioCAD protocol using provided STL files from Nueva Galimplant (Spain). After converting the implant's geometry into a solid, including surface and polysurface union, the models consisted of a fixation cylinder, prosthetic screw, titanium base, and prosthetic restoration [6]. The final models were exported as STEP files after thorough verification (Figure 2).”

The solver and analysis information can be found in the sequence “Each previously described geometry was exported to the computer-aided engineering software (ANSYS 19.2, ANSYS Inc., Houston, TX, USA). A static structural analysis was conducted to calculate the results. In the mechanical module, material information was assigned to each solid component as isotropic and homogeneous, utilizing elasticity modulus and Poisson ratio from previous studies (Table 2). Two different abutment de-sign have been considered similar to the in-vitro test (Figure 3).”

More information about element type, size, mesh density and boundary conditions have been added to the text.

Please also tell me how the convergence of the numerical model was determined - as it was tested - this is also not in the paper.

The convergence information was presented in the sentence “An initial division with tetrahedral elements was automatically generated. Subsequently, a 10% convergence test was employed to assist in mesh refinement and control, ensuring minimal influence on the results of the mathematical calculations [6].”

The great deficiencies in the paper suggest that it should be rejected, but I believe that the authors should be given a chance to revise it and resubmit it for review. I suggest a major revision.

Thank you sincerely for taking the time to provide your valuable comments and review. Your insights are valuable to us, and we greatly appreciate your thoughtful feedback. Once again, we extend our heartfelt gratitude for your contribution to our ongoing efforts.

Reviewer 3 Report

This work researches that the mechanical behavior of CAD/CAM titanium bases used for implant-supported pros-theses can vary considerably. This study aimed to evaluate the impact of antirotational titanium bases on the marginal fit, fatigue behavior, stress concentration, and fracture load of implant-sup-ported CAD/CAM zirconia crowns. Overall, this is a well conducted and easy-to-read study, which adds to the body of knowledge available in this area of research, although major revisions as per the following report should be performed for both technical and language clarity, before the next submission. The comments are below:

1. The current version of the Introduction part is flawed. In the introduction part, the authors need to explain and emphasize the scientific novelty of this work with a comprehensive literature review on the topic. What’s the contribution of the present study to the current research area?

2. The introduction of the low fracture resistance of zirconia ceramics in the preface should include appropriate references to confirm this viewpoint. The authors are encouraged to add some recent publications recommended in the reference section. Such as, W.W. Huang, H.J. Qiu, Y.Q. Zhang, et al. Microstructure and phase transformation behavior of Al2O3-ZrO2 under microwave sintering. Ceramics International, 2022, 49 (03):4855-4862. https://doi.org/10.1016/j.ceramint.2022.09.376. 

3. The statement in the conclusion section is too concise.

4. It is suggested to adjust the alignment of Table 5 to make it more beautiful.

5. References: please carefully read the Submission guidelines of the Journal, and revise the format and keep in consistent. For example, distinguish between uppercase and lowercase letters and the form of giving the number of authors.

I hope these suggestions could be well accepted by the authors.

This work researches that the mechanical behavior of CAD/CAM titanium bases used for implant-supported pros-theses can vary considerably. This study aimed to evaluate the impact of antirotational titanium bases on the marginal fit, fatigue behavior, stress concentration, and fracture load of implant-sup-ported CAD/CAM zirconia crowns. Overall, this is a well conducted and easy-to-read study, which adds to the body of knowledge available in this area of research, although major revisions as per the following report should be performed for both technical and language clarity, before the next submission. The comments are below:

1. The current version of the Introduction part is flawed. In the introduction part, the authors need to explain and emphasize the scientific novelty of this work with a comprehensive literature review on the topic. What’s the contribution of the present study to the current research area?

2. The introduction of the low fracture resistance of zirconia ceramics in the preface should include appropriate references to confirm this viewpoint. The authors are encouraged to add some recent publications recommended in the reference section: (a) W.W. Huang, H.J. Qiu, Y.Q. Zhang, et al. Microstructure and phase transformation behavior of Al2O3-ZrO2 under microwave sintering. Ceramics International, 2022, 49 (03):4855-4862. https://doi.org/10.1016/j.ceramint.2022.09.376. (b)H.Q. Qiu, Y.Q. Zhang, W.W Huang, et al. Sintering properties of tetragonal zirconia nanopowder preparation of the NaCl+KCl binary system by the sol-gel-flux method. ACS Sustainable Chemistry & Engineering, 11, 3, (2023): 1067-1077. https://doi.org/10.1021/acssuschemeng.2c05908.

3. The statement in the conclusion section is too concise.

4. It is suggested to adjust the alignment of Table 5 to make it more beautiful.

5. References: please carefully read the Submission guidelines of the Journal, and revise the format and keep in consistent. For example, distinguish between uppercase and lowercase letters and the form of giving the number of authors.

I hope these suggestions could be well accepted by the authors.

Author Response

This work researches that the mechanical behavior of CAD/CAM titanium bases used for implant-supported pros-theses can vary considerably. This study aimed to evaluate the impact of antirotational titanium bases on the marginal fit, fatigue behavior, stress concentration, and fracture load of implant-sup-ported CAD/CAM zirconia crowns. Overall, this is a well conducted and easy-to-read study, which adds to the body of knowledge available in this area of research, although major revisions as per the following report should be performed for both technical and language clarity, before the next submission. The comments are below:

  1. The current version of the Introduction part is flawed. In the introduction part, the authors need to explain and emphasize the scientific novelty of this work with a comprehensive literature review on the topic. What’s the contribution of the present study to the current research area?

The final paragraph of introduction section contains “However, how the geometrical morphology of the titanium base including antirotational and conical designs affects the stability of the implant-abutment connection was not in-vestigated yet”. Therefore, this can be considered the contribution of the present study to the current research area of materials in dentistry.

  1. The introduction ofthe low fracture resistance of zirconia ceramics in the preface should include appropriate references to confirm this viewpoint. The authors are encouraged to add some recent publications recommended in the reference section: (a) W.W. Huang, H.J. Qiu, Y.Q. Zhang, et al. Microstructure and phase transformation behavior of Al2O3-ZrO2 under microwave sintering. Ceramics International, 2022, 49 (03):4855-4862. https://doi.org/10.1016/j.ceramint.2022.09.376. (b)H.Q. Qiu, Y.Q. Zhang, W.W Huang, et al. Sintering properties of tetragonal zirconia nanopowder preparation of the NaCl+KCl binary system by the sol-gel-flux method. ACS Sustainable Chemistry & Engineering, 11, 3, (2023): 1067-1077. https://doi.org/10.1021/acssuschemeng.2c05908.

Thank you for providing references. The information and insights they offer greatly contribute to the understanding of zirconia's diverse applications and challenges. The introduction has been updated as requested.

  1. The statement in the conclusion section is too concise.

Conclusion section has been improved.

  1. It is suggested to adjust the alignment of Table 5 to make it more beautiful.

Table 5 has been adjusted as requested.

  1. References: please carefully read the Submission guidelines of the Journal, and revise the format and keep in consistent. For example, distinguish between uppercase and lowercase letters and the form of giving the number of authors.

Thank you. We have checked the guidelines and references.

I hope these suggestions could be well accepted by the authors.

Thank you sincerely for your valuable comments and feedback. Your input is greatly appreciated and will undoubtedly contribute to the improvement of our work. We are grateful for your time and effort in providing us with your insightful perspectives.

Round 2

Reviewer 2 Report

I have the impression that the authors ignored my previous comments.

The changes introduced by the authors do not reflect the desire to improve the manuscript - the paper still has the same deficiencies as before sending it back for re-review. They address my objections, but do nothing to raise the substantive level of the manuscript.

I would like to ask the Authors to read the previous review, introduce any corrections, add and correct tables, add figures, and comment on the figures.

Only then can the manuscript be submitted for re-review.

Having read the new version of the paper, I stand by my previous opinion - Major revision".

For your convenience, I am attaching my previous review.

Please make any corrections I require and then submit your manuscript for re-review.

I suggest a major revision.

-------------

Please condense the abstract - in my opinion, it should be concise and indicate what will be in the manuscript, without unnecessary details - this is what the paper is supposed to be about.

The research material used in the laboratory should be called "specimen" in the paper - not "sample" - please correct the manuscript in this respect.

Please add nomenclature to the manuscript - a full list of symbols, abbreviations and markings. It may be at the end of the paper.

Short but to the point introduction.

Table 1 should be improved - more condensed and better formatted.

The paper clearly lacks technical drawings of the specimens used. Please show full isometric views, technical drawings with dimensions, information about the roughness of the speciemns - as we know it affects the results of fatigue tests - roughness affects the fatigue strength. The paper must necessarily include information about the specimens, technical drawings, dimensions of the specimens, illustrations with isometric views of the CAD models of the specimens used.

The manuscript should specify exactly how many specimens were subjected to fatigue tests, what signals were recorded, what these signals were used for. It is worth illustrating these studies with sample charts. In these tests, at least 5 specimens should be used - as it is known, fatigue tests are subject to a significant dispersion - scater of the results. It is worth showing identical graphs from fatigue tests - forces as a function of displacement or time for several specimens, in a specific period of time. In this regard, you should add some results, show their dispersion, perform an analysis.

Tables in the manuscript should be better presented, formatted and discussed.

In addition to photos of broken bolts, it is worth showing and measuring fractures using a laboratory microscope. Please complete your manuscript in this regard.

There is absolutely no information about FEM modeling in the manuscript. Here you should specify the solver, the finite elements used, the number of nodes and integration points in the finite element, the type of interpolation in the finite element, show the FEM model with assumed boundary conditions, external load, etc. Please indicate the material model used in the calculations, show its stress - strain curve give its mechanical parameters. It is necessary to show what type of finite elements were, what was their size - also relate it to the geometric parameter of the specimens, etc. In this respect, the manuscript does not meet any requirements - it disqualifies it from publication.

Please also tell me how the convergence of the numerical model was determined - as it was tested - this is also not in the paper.

The great deficiencies in the paper suggest that it should be rejected, but I believe that the authors should be given a chance to revise it and resubmit it for review. I suggest a major revision.

Minor editing of English language required.

Author Response

I have the impression that the authors ignored my previous comments.

Thank you for expressing your concern. We sincerely apologize if it appeared that your previous comments were ignored. Please be assured that was not our intention, and we greatly value your feedback and contributions to this research. We kindly request that you provide us with the specific comments you feel were overlooked, and we will give them the attention they deserve.

The changes introduced by the authors do not reflect the desire to improve the manuscript - the paper still has the same deficiencies as before sending it back for re-review. They address my objections, but do nothing to raise the substantive level of the manuscript.

I would like to ask the Authors to read the previous review, introduce any corrections, add and correct tables, add figures, and comment on the figures.

Only then can the manuscript be submitted for re-review.

Having read the new version of the paper, I stand by my previous opinion - Major revision".

For your convenience, I am attaching my previous review.

Please make any corrections I require and then submit your manuscript for re-review.

I suggest a major revision.

We would like to reiterate to you that we have addressed the concerns and implemented the modifications as requested in your previous feedback. We understand the importance of providing clear and concise explanations, so we have taken great care to ensure that the revised manuscript reflects these improvements. Below, we outline the specific changes made:

  1. Standardized Terminology: Throughout the manuscript, we have now consistently used the term "specimen" to refer to the experimental samples. 

  2. Improved Table 1: In response to your suggestions, we have thoroughly revised Table 1 to enhance its presentation and readability. 

  3. Isometric Views of Models: To provide a comprehensive understanding of the structural geometry, we have incorporated isometric views of both models. These visual representations offer readers a clear perspective of the design and layout of the models, enabling them to better grasp the intricacies of the research.

  4. Load/Unloading Graph of Fatigue: A load/unloading graph illustrating the fatigue behavior has been added to the manuscript. By including this graph, we aim to enhance the understanding of the experimental findings.

  5. Enhanced Mesh Description: We have made significant improvements to the description of the finite element mesh used in the analysis. The revised section now provides a more comprehensive account of the mesh, including details on the element types employed, element sizes, and any specific considerations taken into account during the mesh generation process.

Regarding your mention of measuring fractures from complex failed specimens, we appreciate your insight. However, upon careful consideration, we have concluded that this approach does not contribute significantly to the present investigation due to the complexity of failed ceramic, present of cement layer and other features that reduces the reliability of such analysis. As such, we have not included measurements of fractures in the manuscript.

After careful consideration, we have decided not to include a list of abbreviations in the document. We understand that such lists can be useful in certain cases, but it is not mandatory for this journal and may not provide significant value to the readers. Moreover, the use of abbreviations in our study is limited and contextually clear, making it unlikely that readers will encounter difficulty in understanding them.

We believe that these modifications have significantly strengthened the manuscript, and we are grateful for your valuable feedback. We hope that these revisions address your concerns and improve the overall quality and clarity of our research. Should you require any further adjustments or have additional suggestions, please do not hesitate to let us know. 

Round 3

Reviewer 2 Report

Once again:

The changes introduced by the authors do not reflect the desire to improve the manuscript - the paper still has the same deficiencies as before sending it back for re-review. They address my objections, but Authors do very little  o nothing to raise the substantive level of the manuscript.

Having read the new version of the paper, I stand by my previous opinion - Major revision".

Please make any corrections I require and then submit your manuscript for re-review.

I suggest a major revision.

Some corrections have been made to the manuscript, but there are still deficiencies - deficiencies in terms of presenting the problem to ordinary engineers.

Please condense the abstract - in my opinion, it should be concise and indicate what will be in the manuscript, without unnecessary details - this is what the paper is supposed to be about.

Please add nomenclature to the manuscript - a full list of symbols, abbreviations and markings. It may be at the end of the paper.

Short but to the point introduction.

Table 1 should be improved - more condensed and better formatted.

!!!! The paper clearly lacks technical drawings of the specimens used. Please show full isometric views, technical drawings with dimensions, information about the roughness of the speciemns - as we know it affects the results of fatigue tests - roughness affects the fatigue strength. The paper must necessarily include information about the specimens, technical drawings, dimensions of the specimens, illustrations with isometric views of the CAD models of the specimens used.

The manuscript should specify exactly how many specimens were subjected to fatigue tests, what signals were recorded, what these signals were used for. It is worth illustrating these studies with sample charts. In these tests, at least 5 specimens should be used - as it is known, fatigue tests are subject to a significant dispersion - scater of the results. It is worth showing identical graphs from fatigue tests - forces as a function of displacement or time for several specimens, in a specific period of time. In this regard, you should add some results, show their dispersion, perform an analysis.

Tables in the manuscript should be better presented, formatted and discussed.

In addition to photos of broken bolts, it is worth showing and measuring fractures using a laboratory microscope. Please complete your manuscript in this regard.

Please also tell me how the convergence of the numerical model was determined - as it was tested - this is also not in the paper.

The great deficiencies in the paper suggest that it should be rejected, but I believe that the authors should be given a chance to revise it and resubmit it for review. I suggest a major revision.

Minor editing of English language required.

Author Response

Dear reviewer,

I am writing to address a concern regarding the dialogue between you as the reviewer and us as the authors during the review process of our manuscript.

Firstly, I want to express our appreciation for the time and effort you have dedicated to reviewing our work. Your feedback and comments have been valuable in improving the quality of our manuscript. However, it has come to our attention that there is a deficiency in the communication between us.

During the previous review rounds, we made sure to thoroughly address all of your queries and comments. We provided detailed responses in a point-by-point letter, carefully explaining the revisions and changes we made to address your concerns. However, it is disheartening to note that in the subsequent review round, we have observed the same sentences and requests being copied and pasted without any acknowledgment of our previous responses. This repetition not only hinders the progress of the review process but also raises concerns about the effectiveness of our communication. It is important to have a meaningful and constructive dialogue to ensure the proper evaluation and improvement of our manuscript. 

We kindly request that moving forward, you take into consideration the responses we have provided in the earlier rounds of revision. 

Thank you for your attention to this concern. We look forward to your continued feedback and a more effective dialogue moving forward.

Best regards,

Authors